# Learning to Be Uncertain: Pre-training World Models with Horizon-Calibrated Uncertainty

**Shenghua, Wan**[1,2]**, Le Gan**[3]**, De-Chuan Zhan**[1,2*]
[1]School of Artificial Intelligence, Nanjing University, Nanjing, China
[2]National Key Laboratory for Novel Software Technology, Nanjing University, Nanjing, China
[3]School of Computer Science and Technology, Nanjing University of Science and
Technology, Nanjing, China
`wansh@lamda.nju.edu.cn,ganle@njust.edu.cn,zhandc@nju.edu.cn`

## Abstract

Pre-training world models on large, action-free video datasets offers a promising path toward generalist agents, but a fundamental flaw undermines this paradigm. Prevailing methods train models to predict a single, deterministic future, an objective that is ill-posed for inherently stochastic environments where actions are unknown. We contend that a world model should instead learn a structured, probabilistic representation of the future where predictive uncertainty correctly scales with the temporal horizon. To achieve this, we introduce a pre-training framework, **H**orizon-c**A**librated **U**ncertainty **W**orld **M**odel (HAUWM), built on a probabilistic ensemble that predicts frames at randomly sampled future horizons. The core of our method is a Horizon-Calibrated Uncertainty (HCU) loss, which explicitly shapes the latent space by encouraging predictive variance to grow as the model projects further into the future. This approach yields a latent dynamics model that is not only predictive but also equipped with a reliable measure of temporal confidence. When fine-tuned for downstream control, our pre-trained model significantly outperforms state-of-the-art methods across a diverse suite of benchmarks, including the DeepMind Control Suite, MetaWorld, and RoboDesk. These results underscore the crucial role of structured uncertainty in informed decision-making.

## 1 Introduction

World models, which learn compressed representations of environmental dynamics, have become a cornerstone of modern reinforcement learning (RL), boosting agent sample efficiency and performance in complex domains (Ha & Schmidhuber, 2018; Hafner et al., 2020; Hu et al., 2023; Wan et al., 2024; Assran et al., 2025; Agarwal et al., 2025). A promising frontier is to pre-train these models on large, action-free video datasets (Seo et al., 2022), allowing them to build a foundational understanding of dynamics from passive observation that can be rapidly fine-tuned for downstream control tasks. This paradigm enables versatile agents to learn without extensive task-specific interaction.

However, a critical flaw undermines current approaches. Video-based pre-training frameworks typically optimize for a single objective: maximizing predictive accuracy under the assumption of a deterministic future (Seo et al., 2022; Wu et al., 2023). This objective is fundamentally misaligned with the nature of action-free video. Without action labels, any observed future is merely one of many possible outcomes, as illustrated in Figure 1. A model trained on this data is erroneously compelled to predict one "correct" future from a state that could unfold in myriad ways.

This deterministic bias creates a paradox. By forcing the model to predict a single future with high precision, pre-training penalizes any representation of environmental stochasticity—the very property that makes real-world dynamics so challenging. Consequently, models learn to suppress

---
*Corresponding Author

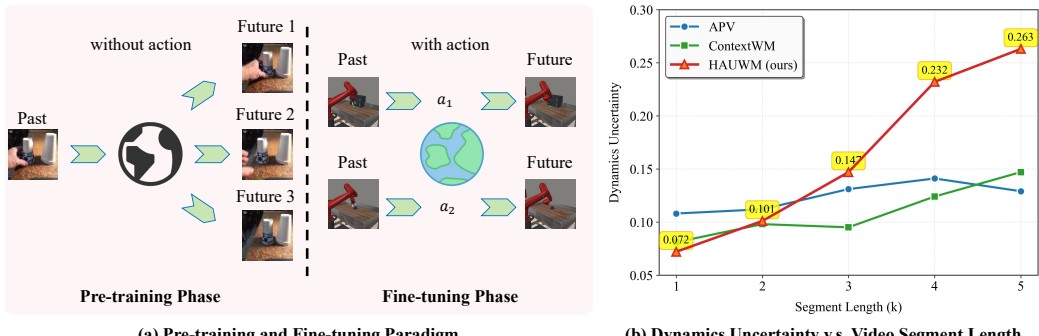

(a) Pre-training and Fine-tuning Paradigm    (b) Dynamics Uncertainty v.s. Video Segment Length

Figure 1: (a) Prevailing pre-training methods erroneously compel a world model to predict a single deterministic outcome from action-free video, ignoring the multiple futures that could unfold from the same past. This deterministic bias creates a fundamental conflict with the fine-tuning stage, in which an agent must learn to use actions to actively select a desired future from among these possibilities. (b) The dynamics uncertainty of various methods pretrained on the video dataset evolves with the temporal interval between predicted frames (the experimental details are in appendix E.3). Notably, HAUWM consistently maintains well-calibrated long-term predictive uncertainty across extended prediction horizons.

ambiguity rather than represent it, fostering a false certainty and losing the capacity for diverse forecasting—as quantitatively evidenced in fig. 1 (b), where standard RSSM-based baselines (APV and ContextWM), even when equipped with ensemble heads, exhibit artificially low and nearly flat predictive uncertainty that fails to grow with the prediction horizon, in stark contrast to the naturally increasing uncertainty of HAUWM. This limitation becomes a significant liability during fine-tuning, when the agent must navigate the action-conditioned dynamics that its pre-trained model was never equipped to handle.

This suggests an opportunity to reframe the objective of action-free pre-training. We contend that instead of prioritizing predictive accuracy alone, a more robust approach is to develop a structured representation of temporal uncertainty. Specifically, world models should learn that predictive confidence decays with the horizon, a principle that is not only under-emphasized but actively suppressed in current methods (Seo et al., 2022; Wu et al., 2023; Gao et al., 2025), leading to the observed uncertainty collapse shown in fig. 1 (b). To realize this, we introduce a new pre-training framework named **H**orizon-c**A**librated **U**ncertainty **W**orld **M**odel (HAUWM) that redefines the model's relationship with the future. Instead of single-step prediction, our method trains a probabilistic ensemble to forecast states across variable, randomly sampled time horizons. The foundation of our approach is the **Horizon-Calibrated Uncertainty (HCU) loss**, which explicitly enforces that the ensemble's predictive variance grows with the prediction horizon.

Our framework produces world models that not only predict future states but also quantify their own confidence across different timescales, a capability vital for robust decision-making. When fine-tuned on benchmarks from the DeepMind Control Suite (Tassa et al., 2018), MetaWorld (Yu et al., 2020a), and RoboDesk (Kannan et al., 2021), our models significantly outperform state-of-the-art methods. This demonstrates that explicitly modeling temporal uncertainty is not just beneficial—it is essential for building truly generalizable agents. Our contributions are:

1. We identify and analyze the key limitation of deterministic prediction in action-free world model pre-training: it suppresses environmental stochasticity rather than representing it.

2. We propose a novel framework using variable-horizon prediction and introduce the Horizon-Calibrated Uncertainty (HCU) loss to learn the relationship between time and predictive uncertainty explicitly.

3. We provide comprehensive empirical evidence that our uncertainty-aware pre-training leads to superior performance across diverse and challenging control benchmarks.

## 2 RELATED WORK

### 2.1 WORLD MODELS FOR REINFORCEMENT LEARNING

Model-based reinforcement learning (MBRL) seeks to improve the sample efficiency of agents by learning a model of the environment's dynamics (Janner et al., 2019; Yu et al., 2020b; 2021). This learned model, often referred to as a "world model", enables an agent to plan or learn behaviors through simulated experience, thereby reducing the need for costly real-world interactions (Sutton, 1991; Kaiser et al., 2020). Early approaches often operated in low-dimensional state spaces; the advent of deep learning enabled the creation of world models that learn directly from high-dimensional sensory inputs, such as images (Janner et al., 2019; Yu et al., 2020b; 2021; Hansen et al., 2024).

A seminal work by Ha & Schmidhuber (2018) demonstrated that a compact, recurrent neural network could capture the essential dynamics of an environment, enabling an agent to solve tasks entirely within its dreamed latent space. This concept was significantly advanced by the development of latent dynamics models, most notably the Recurrent State-Space Model (RSSM) introduced in PlaNet (Hafner et al., 2019) and its successors, the Dreamer family of agents (Hafner et al., 2020; 2021; 2025). These models learn a probabilistic representation of the environment state by combining a deterministic recurrent pathway with a stochastic latent variable. This structure allows them to model complex dynamics and uncertainty while remaining computationally tractable. The policy is then trained efficiently on imagined trajectories generated by rolling out the learned latent dynamics model. While highly effective, the success of these models hinges on their ability to accurately capture the actual environmental dynamics from scratch, a process that can still require substantial in-domain experience, especially in visually complex or diverse settings (Fu et al., 2021; Wan et al., 2023). Our work leverages the power of these latent dynamics models but alleviates the burden of learning from scratch by pre-training on external data.

### 2.2 PRE-TRAINING ON VIDEO DATASETS

The paradigm of pre-training on large, unlabeled datasets has revolutionized fields like natural language processing (Devlin et al., 2019; Radford et al., 2018; Brown et al., 2020) and computer vision (Murahari et al., 2020; He et al., 2022), where models first learn general-purpose representations that are later fine-tuned for specific tasks. Recently, this approach has gained significant traction in reinforcement learning, with a focus on using vast, readily available video data as a source of prior knowledge about the physical world (Seo et al., 2022; Wu et al., 2023; Zhang et al., 2024; Wu et al., 2024; Gao et al., 2025). The core idea is that by observing a wide range of dynamic scenes, a model can learn a foundational understanding of physics, object interactions, and temporal coherence before taking an action in a target environment.

Several works have explored this direction. Action-Free Pre-training from Videos (APV) (Seo et al., 2022) demonstrated that a world model pre-trained on a collection of action-free robotic videos could be effectively fine-tuned for new, unseen manipulation and locomotion tasks. To bridge the gap between the action-free pre-training phase and the action-conditioned fine-tuning phase, APV introduced a stacked architecture that preserves the pre-trained representations while integrating action inputs. Other approaches, such as ContextWM (Wu et al., 2023) and iVideoGPT (Wu et al., 2024), have focused on disentangling static and contextual information from dynamic information in complex in-the-wild videos to improve knowledge transfer. Similarly, some research has explored using contrastive learning objectives on video data to learn useful representations for downstream control (Nair et al., 2022; Xiao et al., 2022). Despite their success, a common limitation unites many of these generative pre-training methods: they typically rely on a deterministic, single-step prediction objective (i.e., predicting frame $t + 1$ from frame $t$). As we argue in this paper, this objective is fundamentally misaligned with the stochastic nature of action-free dynamics (Babaeizadeh et al., 2018), where multiple plausible futures exist. This forces the model to average over all possibilities, often resulting in blurry predictions and a failure to capture the multi-modal nature of the future. Our work directly addresses this limitation by designing a framework that not only predicts the future but also explicitly models and structures its uncertainty over time.

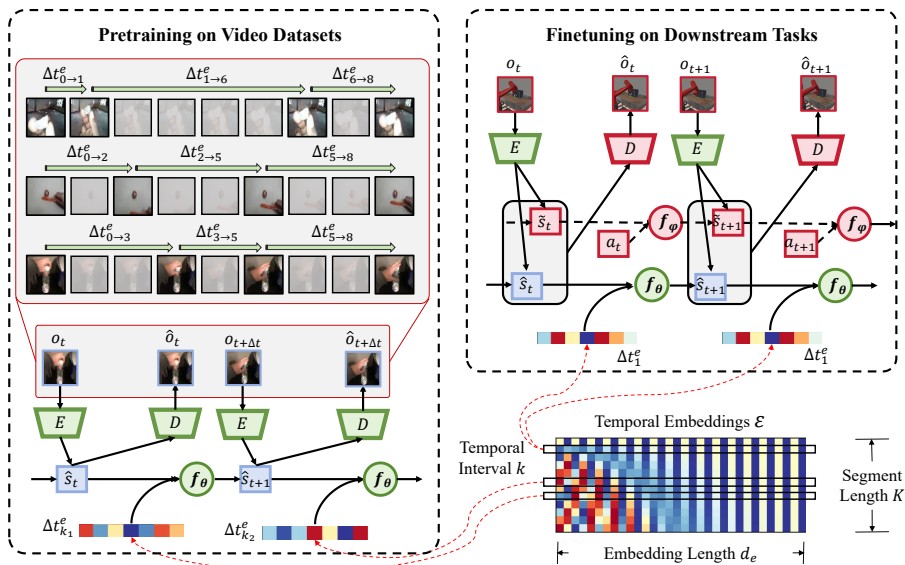

Figure 2: Our pre-training framework (left) uses a dynamics ensemble to predict states at variable horizons, conditioned by a temporal embedding, learning a structured representation of future uncertainty via our HCU loss. During fine-tuning (right), this pre-trained model is frozen to provide an action-agnostic foundation, while a new, lightweight module is trained to learn task-specific, action-conditioned dynamics.

## 3 PRELIMINARIES

Recent work in model-based reinforcement learning has established a powerful two-phase paradigm: unsupervised pre-training on large video datasets followed by task-specific fine-tuning (Seo et al., 2022; Wu et al., 2023; 2024). This approach, exemplified by Action-Free Pre-training from Videos (APV) (Seo et al., 2022), aims to learn generalizable world knowledge from passive, action-free videos, which can then be leveraged to significantly improve the sample efficiency and performance on downstream control tasks (Seo et al., 2022; Wu et al., 2023; 2024).

The first phase involves *action-free pre-training*. Here, a latent video prediction model is trained on a diverse corpus of videos without corresponding action labels. This model is typically a variant of a latent dynamics model, such as a Recurrent State-Space Model (RSSM) (Hafner et al., 2019), which learns to predict future frames by operating in a compressed latent space. It consists of three core components: (i) a representation model $z_t \sim q_\phi(z_t|z_{t-1}, o_t)$ that encodes the current observation $o_t$ into a latent state $z_t$, (ii) a latent transition model $\hat{z}_t \sim p_\phi(\hat{z}_t|z_{t-1})$ that predicts the next state without access to the observation, and (iii) an image decoder $\hat{o}_t \sim p_\phi(\hat{o}_t|z_t)$ that reconstructs the observation from the latent state (Hafner et al., 2019; 2020). The entire model is trained to reconstruct the video sequence by optimizing the variational lower bound (ELBO).

The second phase is *action-conditioned fine-tuning*. To adapt the pre-trained model for a specific RL task, agent actions must be incorporated. A key challenge is to integrate this new information without overwriting the valuable priors learned during pre-training. APV addresses this by introducing a *stacked latent prediction model*, where a new action-conditional dynamics model is stacked on top of the frozen or gently-tuned action-free model. This new model layer produces a task-specific state $s_t$ conditioned on both the agent's action $a_{t-1}$ and the underlying action-free state $z_t$. This composite world model is then used to learn a policy, typically via an actor-critic algorithm that trains on imagined trajectories generated by the model.

## 4 METHODOLOGY

Our methodology is structured around a two-phase framework as shown in fig. 2: (1) uncertainty-aware pre-training on action-free video, followed by (2) efficient fine-tuning for downstream con-

trol tasks. In the pre-training phase, we introduce a variable-horizon prediction task, where an ensemble of dynamics models learns to forecast future states over randomly sampled time intervals. The cornerstone of this phase is our novel Horizon-Calibrated Uncertainty (HCU) loss, which explicitly encourages the predictive variance of the ensemble to increase with the temporal horizon. This procedure endows the model with a structured understanding of temporal uncertainty. For fine-tuning, we freeze the pre-trained uncertainty-aware model and augment it with a lightweight, action-conditioned dynamics module, enabling rapid adaptation to specific control objectives while retaining the foundational world knowledge.

## 4.1 UNCERTAINTY-AWARE PRE-TRAINING ON ACTION-FREE VIDEO SETS

Conventional video-based world model pre-training often erroneously assumes deterministic futures by optimizing single-step prediction accuracy (Seo et al., 2022; Wu et al., 2023). However, video data inherently lacks explicit action labels and permits multiple plausible futures from identical states. This deterministic bias suppresses environmental stochasticity and compromises the model's ability to predict diverse futures—a critical capability for robust decision-making. To resolve this tension, our pre-training framework simultaneously optimizes three key objectives: (1) learning meaningful visual representations from unlabeled video data, (2) maintaining predictive fidelity to observed trajectories, and (3) preserving the capacity to represent multiple plausible futures.

We achieve these objectives using an ensemble of $M$ independent dynamics heads. Each head $p_\theta^{(i)}$ models the stochastic transition to a future latent state $s_{t+k}$, conditioned on the current latent state $s_t$ and temporal horizon $\Delta t_k^e$ (for the consistency of expression, the variable-horizon $k$ and temporal embedding $\Delta t_k^e$ are detailed in section 4.2). Formally, each ensemble head outputs the Gaussian parameters for the future latent state:

$$p_\theta^{(i)}(s_{t+k} \mid s_t, \Delta t_k^e) = \mathcal{N}\left(\mu_{\theta_i}(s_t, \Delta t_k^e), \sigma_{\theta_i}^2(s_t, \Delta t_k^e)I\right) \tag{1}$$

To ensure effective extraction of observational information, we employ a widely used encoder-decoder architecture of Dreamer style (Hafner et al., 2020; Seo et al., 2022). The encoder $q_\phi(s_{t+k} \mid s_t, \Delta t_k^e, o_{t+k})$ infers the latent state $s_{t+k}$ from $s_t$, $\Delta t_k^e$, and future observation $o_{t+k}$. Under POMDP assumptions, the image decoder $p_\phi$ acts as an emission function, reconstructing observations from latent states. The set of predicted means $\{\mu_{\theta_i}\}_{i=1}^M$ represents a discrete distribution over plausible futures. For image reconstruction, we compute the ensemble mean $\bar{\mu}_{t+k} = \frac{1}{M}\sum_{i=1}^M \mu_{\theta_i}(s_t, \Delta t_k^e)$ as $s_{t+k}$ and decode it as $\hat{o}_{t+k} \sim p_\phi(s_{t+k})$.

We optimize model parameters by minimizing a combined objective. The primary predictive loss is the negative variational lower bound:

$$\mathcal{L}_{\text{pred}} = \beta D_{\text{KL}}\left[q_\phi(s_{t+k}|s_t, \Delta t_k^e, o_{t+k}) \,\middle\|\, p_\theta(\hat{s}_{t+k}|s_t, \Delta t_k^e)\right] - \ln p_\phi(\hat{o}_{t+k}|s_{t+k}) \tag{2}$$

where $\beta$ weights the KL-divergence term. This loss forces the model to learn effective representations while maintaining predictive accuracy.

To explicitly preserve future-state diversity, we introduce a **Horizon-Calibrated Uncertainty (HCU) loss**. This loss encourages the ensemble's predictive dispersion to grow monotonically with the temporal horizon $\Delta t_k^e$. To implement this, we adopt a widely-used approach based on the model disagreement (Pathak et al., 2019; Sekar et al., 2020; Seyde et al., 2021):

$$\mathcal{L}_{\text{HCU}} = -k\frac{1}{M-1}\sum_{i=1}^M \left(\mu_{\theta_i}(s_t, \Delta t_k^e) - \bar{\mu}_{t+k}\right)^2 \tag{3}$$

The HCU loss maximizes the model disagreement scaled by horizon length $k$. Minimizing $\mathcal{L}_{\text{HCU}}$ explicitly encodes the inductive bias that uncertainty should increase with prediction horizon.

To dynamically balance predictive accuracy against uncertainty representation, we formulate a dual optimization objective:

$$\mathcal{L}_{\text{total}} = \mathcal{L}_{\text{pred}} + \lambda\mathcal{L}_{\text{HCU}} \tag{4}$$

Here, $\lambda$ serves as an adaptive counterweight that maintains the essential tension between two competing objectives: minimizing reconstruction error ($\mathcal{L}_{\text{pred}}$) to retain predictive fidelity and maximizing uncertainty diversity ($\mathcal{L}_{\text{HCU}}$) to capture environmental stochasticity simultaneously. This self-regulating equilibrium allows the model to maintain high reconstruction quality while progressively

capturing necessary stochasticity—avoiding degenerate solutions where either extreme uncertainty or artificial determinism would degrade downstream performance.

## 4.2 VARIABLE-HORIZON PREDICTION AND RELATIVE TEMPORAL EMBEDDING

Conventional video-based world models predominantly focus on single-step next-frame prediction, neglecting the inherent multi-scale temporal relationships in unlabeled video data (Hong et al., 2022; Brooks et al., 2024). To make visual states exhibit predictable transitions across arbitrary time intervals when equipped with explicit temporal conditioning, we implement variable-horizon prediction during pre-training. For each training instance, we sample a random prediction horizon $k \sim \text{Uniform}\{1, 2, \ldots, K_{\max}\}$ and construct observation pairs $(o_t, o_{t+k})$. To condition the dynamics model on the relative temporal relationship, we generate sinusoidal positional embeddings (Vaswani et al., 2017) for the entire video segment $\mathcal{E} \in \mathbb{R}^{T \times d_e}$, where $T$ is the segment length. We then extract the specific temporal embedding $\Delta t_k^e = \mathcal{E}[k] \in \mathbb{R}^{d_e}$ corresponding to horizon $k$. This relative encoding strategy provides two critical advantages: it inherently normalizes temporal relationships across videos of varying lengths by representing $k$ as a proportion of $T$, and preserves event dynamics by encoding the relative position $k/T$ rather than absolute time steps.

The embedding $\Delta t_k^e$ thereby captures whether $k$ represents a short or long interval relative to the video context, enabling the model to distinguish between rapid and gradual state transitions. This temporal conditioning is input to each ensemble dynamics head, allowing unified modeling of transitions across diverse timescales.Unlike standard positional encodings in Transformers (Vaswani et al., 2017), which primarily index absolute sequence positions to preserve order, our relative temporal embedding $\Delta t_k^e$ explicitly encodes the physical time gap itself. This forces each dynamics head to directly learn the transition $s_t \rightarrow s_{t+k}$ in a single forward pass, rather than relying on recursive single-step rollouts. More critically, this design is tightly coupled with the Horizon-Calibrated Uncertainty (HCU) loss: the embedding serves as a direct conditioning signal for the expected degree of stochasticity, enabling the HCU loss to reliably enforce monotonic variance growth with horizon length—a mechanism absent in prior temporal conditioning approaches that target only mean accuracy. Thus, variable-horizon prediction is not merely a positional encoding variant, but an essential architectural choice deliberately engineered to make structured, horizon-dependent uncertainty emerge naturally and controllably. Implementation details are provided in appendix C.2.

## 4.3 FINE-TUNING FOR DOWNSTREAM CONTROL

Having established an uncertainty-aware world model through action-free video pre-training, we adapt this foundation to task-specific control via a structured fine-tuning phase. This transition preserves the rich dynamics and stochasticity learned during pre-training while integrating action-conditioned transitions essential for decision-making. We achieve this by augmenting the pre-trained model with a dedicated task-oriented pathway, following established pretraining-finetuning paradigms (Seo et al., 2022; Wu et al., 2023) but critically extending them to maintain uncertainty calibration. Specifically, we initialize the visual encoder and ensemble dynamics heads from pre-training, freezing their parameters to retain general visual and temporal representations, while introducing a new action-conditioned dynamics stream trained from scratch.

Our fine-tuning architecture processes two parallel latent streams. The pre-trained stream—unmodified from pre-training—receives only temporal embeddings and generates latent states $\hat{s}_t$ that capture action-agnostic environmental dynamics. To maintain compatibility with this stream's original design, we condition it on relative temporal embeddings $\Delta t_{k=1}^e$ (as defined in Section 4.2), injecting a Gaussian noise ($\sigma = 0.01$) to enhance robustness against temporal discretization artifacts. Concurrently, a new action-conditioned stream processes agent actions $a_{t-1}$ and the pre-trained latent state $\hat{s}_t$ to produce task-specific predictions $\tilde{s}_t$ via $p_\psi(\tilde{s}_t \mid s_{t-1}, a_{t-1}, \hat{s}_t)$. This dual-stream design ensures that the model leverages pre-trained uncertainty awareness while learning real action effects. The composite latent state $[\hat{s}_t; \tilde{s}_t]$ then drives policy optimization, with an auxiliary reward predictor $R_\theta$ trained on task-specific rewards to guide imagination-based planning.

Using this integrated world model, we execute standard model-based reinforcement learning: the agent learns an actor-critic policy by generating imagined trajectories entirely within the calibrated latent space. Crucially, the pre-trained ensemble dynamics provide uncertainty-aware state predictions during imagination, enabling the policy to account for environmental stochasticity while

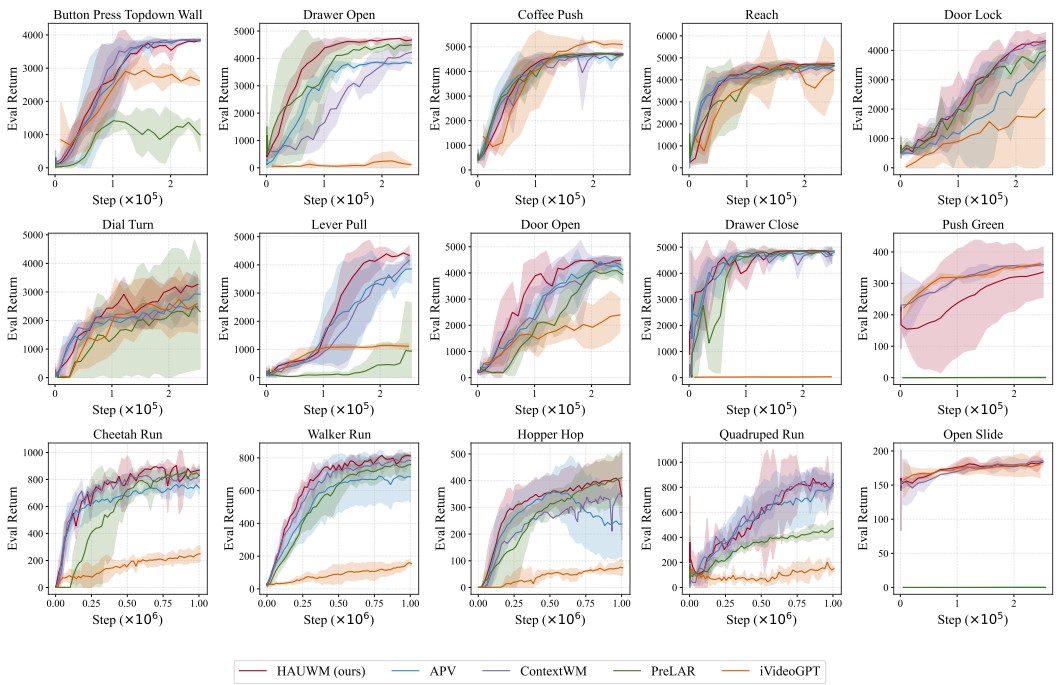

Figure 3: Performance comparison of HAUWM against state-of-the-art baselines on a suite of downstream manipulation and locomotion tasks. Solid curves represent the mean evaluation return across four random seeds, and shaded regions denote the 95% confidence interval.

exploiting task-specific action knowledge from the fine-tuned stream. The overall algorithm is in appendix B, and full architectural details appear in appendix C.1.

## 5 EXPERIMENTS

We conduct a series of experiments to validate our proposed framework and answer the following key research questions:

1. **Performance (RQ1):** Does HAUWM lead to improved sample efficiency and final performance on downstream RL tasks compared to state-of-the-art methods?

2. **Ablation (RQ2):** What are the relative contributions of the core components of HAUWM?

3. **Analysis (RQ3):** Does HAUWM successfully estimate the uncertainty in different training stage?

4. **Robustness (RQ4):** Can our pre-training world model generalize to diverse downstream learning paradigms?

### 5.1 EXPERIMENTAL SETUP

**Benchmark Environments.** We evaluate our method on a diverse suite of challenging, vision-based continuous control benchmarks. These include several locomotion tasks from the **DeepMind Control Suite (DMC)** (Tassa et al., 2018), a set of distinct robotic manipulation tasks from **Meta-World** (Yu et al., 2020a), and complex, long-horizon tasks from **RoboDesk** (Kannan et al., 2021). These environments test the agent's ability to learn complex motor skills, generalize across tasks, and operate from high-dimensional pixel inputs. All observations are rendered as $64 \times 64 \times 3$ images.

**Pre-training Data.** For the pre-training phase, we utilize a large, publicly available video dataset, Something-Something-v2 (Goyal et al., 2017), for HAUWM and all baselines. The key charac-

teristic is that these videos are action-free and sourced from domains that may differ visually and dynamically from the downstream fine-tuning tasks. More details are in appendix A.

**Baselines.** We compare our method, which we denote as **HAUWM**, against several strong baselines to provide a comprehensive evaluation:

- **APV** (Seo et al., 2022): A leading method that first establishes a foundational framework for leveraging unlabeled videos through stacked latent prediction.
- **ContextWM** (Wu et al., 2023): A state of the art method that effectively handles complex in-the-wild videos by disentangling contextual and dynamic information.
- **PreLAR** (Zhang et al., 2024): A method that learns implicit action representations directly from observation pairs during pre-training and fine-tuning with real actions.
- **iVideoGPT** (Wu et al., 2024): A flexible framework that shows scalability through transformer-based architectures trained on over one million manipulation trajectories.

## 5.2 DOWNSTREAM TASK PERFORMANCE (RQ1)

As illustrated in fig. 3, our method, HAUWM, achieves state-of-the-art sample efficiency and final performance on the majority of the tested benchmarks. The advantage is particularly pronounced in dynamically complex locomotion tasks such as *Walker Run*, and *Hopper Hop*. In these environments, HAUWM consistently learns faster and converges to a higher final return than all baselines. We attribute this strong performance to our core contribution: by pre-training a model that explicitly represents temporal uncertainty, the agent builds a more robust and realistic internal model of the world. This foundation allows it to adapt more effectively during fine-tuning, where it must control its actions amidst environmental stochasticity.

Table 1: Ablation study of HAUWM.

| Method | DMC | MetaWorld | RoboDesk |
|---|---|---|---|
| $\lambda = 10.0$ | 0.67±0.13 | 0.77±0.05 | 0.61 ± 0.09 |
| $\lambda = 10^{-1}$ | 0.69±0.06 | 0.80±0.10 | 0.60 ± 0.05 |
| $\lambda = 10^{-2}$ | 0.70±0.04 | 0.76±0.07 | 0.65 ± 0.07 |
| w/o HCU | 0.64±0.11 | 0.73±0.14 | 0.55 ± 0.08 |
| **HAUWM** | **0.74±0.03** | **0.85±0.05** | **0.71 ± 0.05** |

Conversely, on the *Push Green* task, HAUWM is outperformed by ContextWM. We hypothesize this is because the task's dynamics are more deterministic and narrowly goal-oriented, reducing the relative benefit of sophisticated uncertainty modeling. In such settings, methods that learn a direct, implicit action-to-outcome mapping may have an advantage. The somewhat larger performance variance of HAUWM on tasks like *Dial Turn* is an expected trade-off of our ensemble-based architecture, which, while enabling superior average performance, can naturally lead to greater diversity across independent training seeds.

## 5.3 ABLATION STUDIES (RQ2)

We conduct a series of ablation studies to dissect the contributions of HAUWM's key components and validate our design choices, and record results in table 1 and fig. 4. Each value is normalized against the random and expert returns and averaged across all corresponding tasks.

First, to verify the necessity of our core contribution, we trained a variant without the Horizon-Calibrated Uncertainty loss (w/o HCU). This led to significant performance degradation, particularly on the DMC benchmark, confirming that explicitly modeling structured temporal uncertainty is critical for learning robust dynamics representations. Next, we examined the sensitivity to the initial weight of HCU loss, $\lambda$. The results indicate that performance suffers with both overly large ($\lambda = 10.0$) and excessively small ($\lambda = 10^{-2}$) values. A large $\lambda$ forces the model to prioritize uncertainty at the expense of predictive fidelity, while a small $\lambda$ provides an insufficient training signal. Our final model employs a moderate weight ($\lambda = 1.0$), striking an effective balance between these competing objectives. The episodic returns of each task are in table 4.

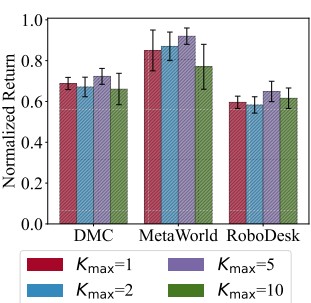

Figure 4: Ablation results on $K_{\max}$.

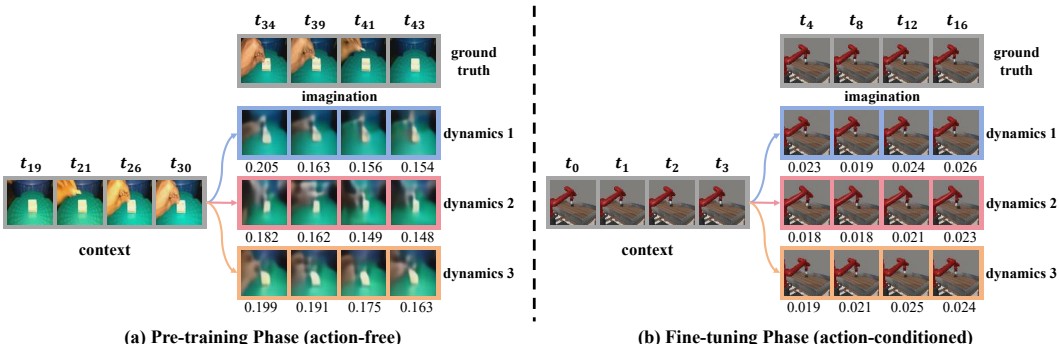

Figure 5: Imagined future rollouts from three randomly selected dynamics heads during action-free pre-training (a) and action-conditioned fine-tuning (b). The model produces diverse, high-uncertainty futures when no action is given, which converge to a single, low-uncertainty outcome when conditioned on a specific action, with uncertainty values noted below each frame.

Finally, we analyzed the impact of the maximum prediction horizon, $K_{max}$. As shown in fig. 4, a moderate value of $K_{max} = 5$ yields the best performance. A small horizon like $K_{max} = 1$ or $2$ provides an insufficient range of temporal intervals to learn a meaningful uncertainty curve. Conversely, a large horizon like $K_{max} = 10$ can make the long-range prediction task overly complex, introducing noisy signals that may destabilize training. For additional ablation studies examining the scaling function $k$ and the number of ensemble dynamics heads $M$, refer to the appendix E.4 and appendix E.5.

## 5.4 ANALYSIS OF LEARNED UNCERTAINTY (RQ3)

To address RQ3, we perform a qualitative analysis of the model's uncertainty estimation capabilities at different stages. During the action-free pre-training phase (left in fig. 5), HAUWM successfully preserves predictive diversity, demonstrating its capacity to represent environmental stochasticity. Different heads of the dynamics ensemble forecast distinct, plausible futures from the same initial state; for instance, dynamics model 1 imagines the robotic arm lifting away from the surface, whereas model 3 predicts it continuing to push rightward. The model correctly learns that uncertainty increases with the prediction horizon, which aligns with our design of the HCU loss. As shown by the quantitative values below the frames, the uncertainty associated with predictions further in the future (e.g., frame predicted at $t_{34}$ from $t_{30}$) is consistently higher than for nearer frames (e.g., frame predicted at $t_{43}$ from $t_{41}$), validating the efficacy of our HCU loss.

During fine-tuning (right in fig. 5), this learned agent actions appropriately constrain stochasticity. When conditioned on a specific and same action sequence, the different dynamics models converge on a single, consistent outcome. Concurrently, the associated uncertainty values drop dramatically compared to the action-free stage. This behavior demonstrates that HAUWM learns a well-calibrated representation of temporal uncertainty, fulfilling our primary motivation.

## 5.5 PERFORMANCES UNDER DIVERSE DOWNSTREAM LEARNING PARADIGMS (RQ4)

A crucial test for a pre-trained foundation model is its versatility across different downstream learning paradigms, not just standard online RL. Therefore, we fine-tune our pre-trained model using established algorithms for imitation learning (IL) and offline RL on several benchmark tasks, with standard VMAIL (Rafailov et al., 2021b) and LOMPO (Rafailov et al., 2021a) algorithms, respectively (extra experimental details are in appendix E). The results, presented in table 2, demonstrate the broad applicability of our approach. In the imitation learning setting, HAUWM achieves state-of-the-art performance, outperforming baselines in several tasks and performing competitively in *Drawer Close*. This suggests that the structured uncertainty learned during pre-training provides a robust foundation for mimicking experts, where understanding plausible future states is crucial.

Furthermore, HAUWM shows strong performance in the offline RL, where learning from a fixed dataset without online interaction is required. It significantly surpasses the baselines on almost all

Table 2: Performance of HAUWM and baselines on diverse downstream learning paradigms, including imitation and offline RL. All scores are normalized returns, reported as the mean and standard deviation across 10 evaluation trajectories (details in appendix E).

|  |  | Cheetah Run | Hopper Hop | Dial Turn | Drawer Close | Push Green |
|---|---|---|---|---|---|---|
| Imitation | APV | $0.78 \pm 0.11$ | $0.83 \pm 0.07$ | $0.55 \pm 0.13$ | $0.89 \pm 0.21$ | $\mathbf{0.71 \pm 0.05}$ |
|  | ContextWM | $0.91 \pm 0.04$ | $0.85 \pm 0.05$ | $0.69 \pm 0.09$ | $\mathbf{0.98 \pm 0.03}$ | $0.69 \pm 0.05$ |
|  | PreLAR | $0.82 \pm 0.07$ | $0.81 \pm 0.11$ | $0.72 \pm 0.06$ | $0.93 \pm 0.05$ | $0.69 \pm 0.05$ |
|  | HAUWM | $\mathbf{0.94 \pm 0.03}$ | $\mathbf{0.89 \pm 0.05}$ | $\mathbf{0.77 \pm 0.06}$ | $0.97 \pm 0.02$ | $0.63 \pm 0.06$ |
| Offline RL | APV | $0.56 \pm 0.06$ | $0.66 \pm 0.12$ | $\mathbf{0.74 \pm 0.13}$ | $0.71 \pm 0.09$ | $0.63 \pm 0.04$ |
|  | ContextWM | $0.61 \pm 0.06$ | $0.69 \pm 0.06$ | $0.69 \pm 0.07$ | $0.70 \pm 0.05$ | $0.66 \pm 0.03$ |
|  | PreLAR | $0.71 \pm 0.08$ | $0.65 \pm 0.05$ | $0.69 \pm 0.02$ | $0.68 \pm 0.03$ | $0.69 \pm 0.09$ |
|  | HAUWM | $\mathbf{0.81 \pm 0.13}$ | $\mathbf{0.71 \pm 0.08}$ | $0.67 \pm 0.12$ | $\mathbf{0.74 \pm 0.07}$ | $\mathbf{0.70 \pm 0.04}$ |

tasks. While APV performs better on the *Dial Turn* task, HAUWM's overall significant performance across both IL and offline RL highlights its versatility. Our uncertainty-aware pre-training strategy equips the model with a greater representational capacity under diverse downstream learning paradigms. This adaptability validates that HAUWM is not just a pre-training solution for online RL but a powerful and versatile foundation model for a wide array of decision-making tasks.

## 6 CONCLUSION

In this work, we address a critical limitation in pre-training world models from action-free video: the reliance on deterministic prediction objectives, which are ill-posed for inherently stochastic environments. We posit that models must instead explicitly learn temporal uncertainty—the principle that long-term futures are less predictable than near-term ones. Our proposed framework achieves this through a variable-horizon prediction task and a novel Horizon-Calibrated Uncertainty (HCU) loss. The HCU loss incentivizes an ensemble's predictions to diverge systematically as the time horizon increases. Across multiple benchmarks, our method achieves state-of-the-art sample efficiency and final performance. This work demonstrates that making structured uncertainty a primary objective of pre-training is crucial for building capable and general-purpose world models.

Future work will focus on developing more computationally-efficient probabilistic models, such as diffusion models (Ho et al., 2020), and improving their scalability with large-scale Transformer architectures (Vaswani et al., 2017; Brown et al., 2020). Scaling our approach to massive, in-the-wild video datasets remains a promising direction for creating truly generalist agents.

### ACKNOWLEDGMENTS

We thank Yucen Wang, Xingye Xu, and Jiabei Lv for their valuable discussions. This work was partially supported by the Young Scientists Fund of the National Natural Science Foundation of China (Ph.D Candidate) under Grant No. 624B200197, the National Science and Technology Major Project under Grant No. 2022ZD0114805, Collaborative Innovation Center of Novel Software Technology and Industrialization, NSFC (62376118, 62006112, 62250069, 61921006), and the "111 Center" (No. B26023).

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

## THE USE OF LARGE LANGUAGE MODELS

In preparing this work, we utilized a Large Language Model (LLM) solely for limited, auxiliary purposes. Specifically, the LLM was employed to assist with refining the English phrasing of certain paragraphs, correcting grammatical errors, and, in the Related Work section, for generating ideas and searching for potentially relevant literature. The model did not contribute to the core research ideation, mathematical formulation, theoretical analysis, algorithm development, experimental design, or interpretation of results. All authors take full responsibility for the entire content of this work, including any text generated with the aid of the LLM.

## A  ENVIROMENTS AND DATASETS

We evaluate our method across three representative benchmarks that collectively span diverse domains of visual reinforcement learning. The DeepMind Control Suite (DMC) (Tassa et al., 2018) provides a standardized testbed for locomotion tasks with physically realistic dynamics, featuring continuous control challenges that require precise motor coordination and long-term planning. Specifically, we examine *walker_run*, which tests bipedal locomotion stability; *cheetah_run*, evaluating high-speed quadrupedal movement; *hopper_hop*, assessing single-legged balance and propulsion; and *quadruped_run*, measuring complex multi-joint coordination for agile movement. Meta-World (Yu et al., 2020a) offers a comprehensive benchmark of robotic manipulation tasks with varying geometric and physical properties, enabling rigorous evaluation of transfer learning capabilities across structurally different but conceptually related challenges. Our evaluation includes *dial_turn*, requiring precise rotational control; *lever_pull*, testing force application at mechanical advantage points; *door_open*, evaluating sequential manipulation of articulated objects; and *drawer_close*, assessing fine-grained spatial reasoning for object containment. Finally, RoboDesk (Kannan et al., 2021) presents real-world inspired manipulation scenarios with increased visual complexity and longer horizons, where *push_green* challenges object relocation with color-based discrimination,

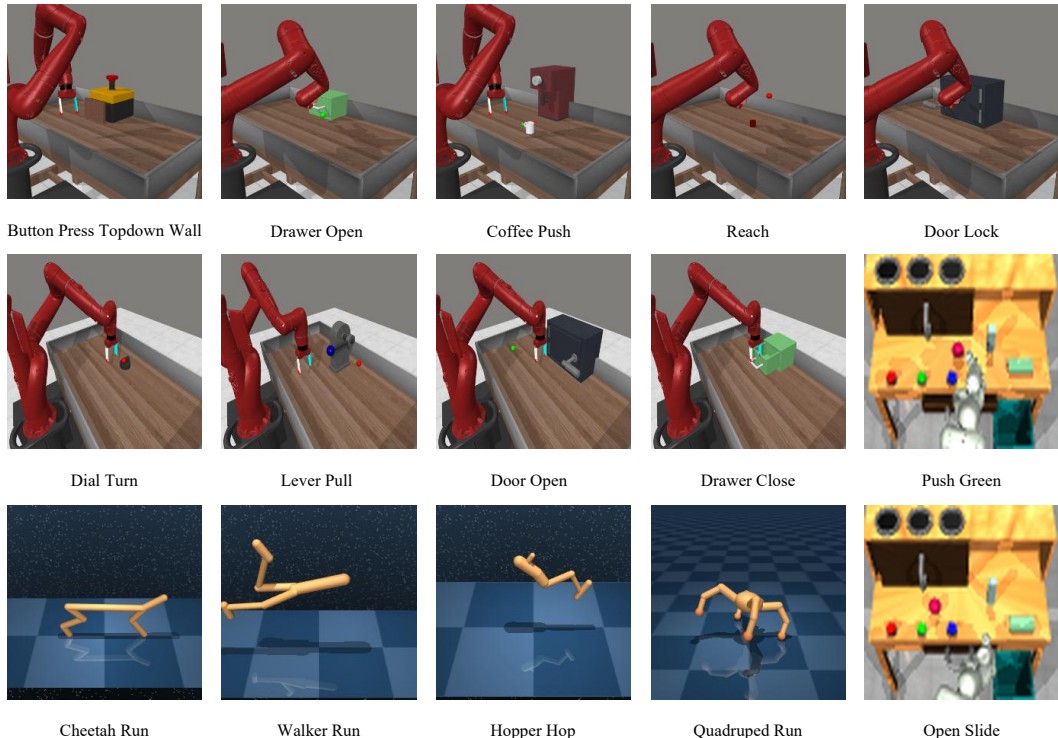

Figure 6: The observation example of each task in the environments: Meta-World (top left), DMC (bottom left), and RoboDesk (right).

while *open_slide* demands precise linear actuation of constrained mechanisms. We display the observation example of each task in fig. 6.

For pre-training, we leverage the Something-Something-V2 (SSv2) dataset (Goyal et al., 2017), a large-scale collection of 220,000 short video clips capturing humans performing basic physical interactions with everyday objects under natural lighting conditions. This dataset provides diverse action demonstrations—including "Putting something on a surface", "Moving something up", and "Covering something with something"—that, despite originating from non-robotic contexts, contain fundamental physical dynamics transferable to robotic manipulation. By utilizing this in-the-wild video resource, our approach demonstrates that meaningful world models can emerge from general human-object interactions rather than domain-specific robotic data, significantly reducing the need for task-specific pre-training collections while maintaining strong transfer performance to downstream robotic control tasks.

## B    PSEUDO CODES

We list the pseudo-codes of HAUWM in algorithm 1.

## C    MODEL DETAILS

### C.1    MODEL ARCHITECTURE

Our model architecture builds upon the Recurrent State-Space Model (RSSM) framework (Hafner et al., 2019; 2020) and is composed of several key modules: a shared image encoder, a corresponding decoder, and dynamics models for the pre-training and fine-tuning phases.

**Encoder and Decoder**    We use a ResNet-based architecture for both the visual encoder and decoder, enabling the processing of $64 \times 64 \times 3$ image observations.

---

**Algorithm 1** Uncertainty-Aware Pre-training and Fine-tuning

---

1: **Input**: Action-free video dataset $\mathcal{D}_{\text{video}}$, task interaction budget
2: **Initialize**: World model parameters $\theta, \phi$; adaptive weight $\lambda$; task replay buffer $\mathcal{D}_{\text{task}} \leftarrow \emptyset$
3:
4: **Phase 1: Uncertainty-aware Pre-training with Dual Optimization**
5: **for** each pre-training step **do**
6:     Sample video clip from $\mathcal{D}_{\text{video}}$ and a horizon $k \sim \text{Uniform}\{1, \ldots, K_{\max}\}$.
7:     Get temporal embedding $\Delta t_k^e$ for horizon $k$.
8:     Compute ensemble predictions $\{\mu_{\theta_i}(s_t, \Delta t_k^e)\}_{i=1}^M$ and model uncertainty.
9:     Calculate total loss $\mathcal{L}_{\text{total}} = \mathcal{L}_{\text{pred}} + \lambda \mathcal{L}_{\text{HCU}}$.
10:     Update world model parameters $\theta, \phi$ by minimizing $\mathcal{L}_{\text{total}}$.
11:     Update $\lambda$ via dual optimization to balance $\mathcal{L}_{\text{pred}}$ and $\mathcal{L}_{\text{HCU}}$.
12: **end for**
13:
14: **Phase 2: Fine-tuning for Downstream Control**
15: Initialize action-conditioned dynamics $p_\psi$ and policy $\pi_\xi$. Freeze $\theta, \phi$.
16: **for** each environment step up to budget **do**
17:     *// World Model Adaptation and Policy Learning*
18:     Sample trajectories from $\mathcal{D}_{\text{task}}$ to update $p_\psi$ and reward predictor $R_\theta$.
19:     Imagine trajectories using the policy $\pi_\xi$ and the composite world model $(p_\theta, p_\psi, R_\theta)$.
20:     Update policy $\pi_\xi$ on imagined trajectories.
21:     *// Environment Interaction*
22:     Execute action $a_t \sim \pi_\xi(s_t)$, observe $(o_{t+1}, r_{t+1})$ and add to $\mathcal{D}_{\text{task}}$.
23: **end for**
24: **return** Trained policy $\pi_\xi$

---

- **Encoder**: The encoder is a convolutional neural network (CNN) with 3 residual blocks. It processes an input image $o_t$ to produce a low-dimensional embedding. This embedding serves as the primary input for the dynamics model.

- **Decoder**: The decoder is a transposed convolutional neural network, also with 3 residual blocks. It reconstructs the image observation $\hat{o}_t$ from a given latent state $s_t$.

**Pre-training**    During the action-free pre-training phase, the world model learns environmental dynamics using a variable-horizon prediction task.

- **Dynamics Model**: We use an ensemble of $M = 5$ dynamics heads built upon an `EnsembleRSSM`. The model features a deterministic state of size 200 and a stochastic state of size 30. It takes the image embedding and a sinusoidal temporal embedding $\Delta t_k^e$ to predict a future latent state $s_{t+k}$.

**Fine-tuning**    For downstream tasks, we freeze the pre-trained components and introduce a new action-conditioned module.

- **Dynamics Model**: The fine-tuning architecture employs a dual-stream design.
  - The frozen, pre-trained `EnsembleRSSM` generates an action-agnostic latent state $\hat{s}_t$ conditioned on the image embedding and a fixed temporal embedding $(\Delta t_{k=1}^e)$.
  - A new, lightweight action-conditioned `RSSM`, trained from scratch, produces a task-specific latent state $\tilde{s}_t$ from the agent's action $a_{t-1}$ and the action-agnostic state $\hat{s}_t$.

- **Actor-Critic and Reward Predictor**: The policy and value functions are MLPs that take the composite latent state $[\hat{s}_t; \tilde{s}_t]$ as input. The reward predictor is a separate MLP that estimates the task reward from the task-specific state $\tilde{s}_t$. Both consist of 2 hidden layers with 400 units each and employ the ELU activation function.

## C.2 TRAINING DETAILS

We implement our framework in PyTorch and use the Adam optimizer for all model components. All experiments are run on NVIDIA A6000 GPUs with 16-bit precision (FP16) to accelerate training and take about 2000 GPU hours.

**Pre-training Phase**  The goal of the pre-training phase is to learn a robust, uncertainty-aware world model from action-free video data.

- **Data and Batching**: We pre-train our model on the Something-Something-v2 (SSv2) dataset (Goyal et al., 2017). To implement our variable-horizon prediction strategy, we process each video into training sequences. For each sample, we extract a sequence of frames with length $L$, where the temporal gap $k$ between any two consecutive frames is randomly sampled from a uniform distribution, $k \sim \text{Uniform}\{1, \ldots, K_{max}\}$. We set the maximum horizon to $K_{max} = 5$. This creates batches where the model must predict across varied and non-contiguous time steps.

- **Temporal Embedding**: To inform the model of the time gap, we generate a relative temporal embedding $\Delta t_k^e$ for each sampled gap $k$. This is achieved by creating a sinusoidal positional encoding matrix for the full length of the source video; the embedding for a specific gap $k$ is then looked up from this matrix. This embedding is supplied as input to the dynamics model, analogous to an action.

- **Optimization**: The world model is trained for 600k gradient steps. We use the Adam optimizer with a learning rate of $1 \times 10^{-4}$ and a batch size of 32. The adaptive weight $\lambda$ for the HCU loss is also optimized with Adam, using a learning rate of $3 \times 10^{-4}$.

**Fine-tuning Phase**  In the fine-tuning phase, the pre-trained model is adapted for downstream control tasks.

- **Initialization**: We initialize the visual encoder and the action-agnostic dynamics model with the weights from the pre-training phase and freeze them. The new action-conditioned dynamics model and the actor-critic policy are trained from scratch.

- **Training Loop**: The agent first collects an initial dataset of 5,000 steps by executing a random policy. It then enters the main training loop, alternating between environment interaction to collect new data into a replay buffer and updating the model parameters. The agent is trained for a total of 1 million steps in the DMC environment and 250 thousand steps in the Meta-World and RoboDesk environments.

- **Temporal Conditioning**: To maintain compatibility with the pre-trained components, the frozen action-agnostic dynamics stream is conditioned on a fixed temporal embedding for $k = 1$, with minor Gaussian noise ($\sigma = 0.01$) added to enhance robustness.

- **Optimization**: The new world model components (action-conditioned stream and reward predictor) are trained with the Adam optimizer using a learning rate of $3 \times 10^{-4}$. The actor and critic components are trained with a learning rate of $8 \times 10^{-5}$.

## C.3 HYPER PARAMETERS

We summarize the key hyperparameters used for the pre-training and fine-tuning phases in table 3.

## D DETAILED BASELINES

**APV** (Action-Free Pre-training from Videos) (Seo et al., 2022) introduces a two-phase paradigm that first pre-trains a world model on unlabeled videos without action information, then adapts it to downstream reinforcement learning tasks through fine-tuning. During pre-training, APV learns a generative video prediction model by optimizing a standard single-step predictive objective on action-free video sequences, effectively capturing visual dynamics without requiring action labels. To bridge the gap between action-free pre-training and action-conditioned reinforcement learning, APV employs a stacked architecture that preserves pre-trained representations while integrating

| Phase | Hyperparameter | Value |
|---|---|---|
| **Pre-training** | Batch Size | 32 |
| | Sequence Length | $len(video)$ |
| | Max Prediction Horizon ($K_{max}$) | 5 |
| | Total Gradient Steps | 600,000 |
| | Ensemble Size ($M$) | 7 |
| | KL Loss Balance | 0.8 |
| | Initial HCU Weight ($\lambda$) | 1.0 |
| | World Model Learning Rate | $1 \times 10^{-4}$ |
| | HCU $\lambda$ Learning Rate | $3 \times 10^{-4}$ |
| | Optimizer | Adam |
| **Fine-tuning** | Batch Size | 32 |
| | Sequence Length | 50 |
| | Replay Buffer Capacity | $1 \times 10^6$ |
| | Initial Random Steps (Prefill) | 10,000 |
| | Total Environment Steps | |
| | DMC | $1 \times 10^6$ |
| | Meta-World & RoboDesk | $2.5 \times 10^5$ |
| | Imagination Horizon | 15 |
| | Discount Factor ($\gamma$) | 0.99 |
| | GAE Parameter ($\lambda_{GAE}$) | 0.95 |
| | World Model Learning Rate | $3 \times 10^{-4}$ |
| | Actor-Critic Learning Rate | $8 \times 10^{-5}$ |
| | Optimizer | Adam |

Table 3: Core hyperparameters for HAUWM pre-training and fine-tuning.

action inputs during fine-tuning. This approach significantly improves sample efficiency in downstream tasks by transferring knowledge from diverse video datasets to specific control problems, establishing an important foundation for world model pre-training that subsequent methods have built upon.

**ContextWM** (Contextualized World Models) (Wu et al., 2023) specifically addresses the challenges of pre-training world models on complex, diverse in-the-wild videos that contain numerous contextual factors irrelevant to task dynamics. Rather than treating all visual information equally, ContextWM explicitly disentangles static contextual elements (such as backgrounds and object appearances) from dynamic state transitions through a specialized architecture. During pre-training, it learns to separate these components, preserving only the dynamics-relevant information while filtering out task-irrelevant variations. This contextual disentanglement enables more effective knowledge transfer when fine-tuning on downstream tasks, particularly when pre-training data comes from visually diverse sources that differ significantly from target environments. By focusing on the core dynamics rather than superficial visual variations, ContextWM achieves superior performance on a wide range of control tasks despite the domain gap between pre-training and target environments.

**PreLAR** (World Model Pre-training with Learnable Action Representation) (Zhang et al., 2024) addresses the fundamental limitation of previous approaches by incorporating action-conditional learning during the pre-training phase, even when explicit action labels are unavailable. Instead of treating pre-training as a pure video prediction task, PreLAR extracts implicit action representations by encoding observations from two consecutive time steps, effectively learning the causal transitions between states from unlabeled videos. The method introduces an action-state consistency loss to ensure these learned representations align with actual dynamics, creating a more seamless transition between pre-training and fine-tuning phases. This innovative approach eliminates the architectural discrepancy between pre-training (video prediction) and fine-tuning (action-conditional world model), resulting in more effective knowledge transfer and significantly improved sample

efficiency across various visual control tasks, particularly in robotic manipulation scenarios where action understanding is critical.

**iVideoGPT** (Interactive VideoGPTs) (Wu et al., 2024) represents a significant advancement in world model scalability by leveraging transformer architectures to handle long video sequences while maintaining interactivity. Unlike RNN-based approaches that process sequences step-by-step, iVideoGPT employs a token-based video prediction framework that processes entire video sequences in parallel, enabling efficient training on massive datasets comprising over one million trajectories from both robotic and human manipulation. The model uses a cross-entropy loss to predict subsequent video tokens autoregressively, creating a versatile foundation that can be adapted to various downstream tasks through domain-specific fine-tuning. Crucially, iVideoGPT demonstrates that world models can be both highly scalable (processing long sequences efficiently) and interactive (supporting action conditioning for control), overcoming a fundamental limitation in previous approaches that often sacrificed one capability for the other. This dual capability makes it particularly effective for complex manipulation tasks requiring both long-term planning and responsive interaction.

# E    EXTRA EXPERIMENTAL DETAILS

## E.1    DIVERSE DOWNSTREAM LEARNING PARADIGMS

To rigorously assess the versatility of our pre-trained world model, we fine-tune it on downstream tasks using two distinct and challenging learning paradigms: model-based imitation learning (IL) and offline reinforcement learning (RL). This tests the model's ability to serve as a robust foundation not only for online exploration but also for learning from expert demonstrations and static, pre-collected datasets.

For the imitation learning paradigm, we adapt the Visual Model-based Adversarial Imitation Learning (VMAIL) framework (Rafailov et al., 2021b). Instead of training a variational dynamics model from scratch as in the original work, we leverage the fine-tuned, action-conditioned world models (APV (Seo et al., 2022), ContextWM (Wu et al., 2023), HAUWM) as the latent environment for policy optimization. Expert demonstrations are first encoded into latent trajectories using the frozen pre-trained encoder. A discriminator, $D_\psi$, is then trained to distinguish these latent expert trajectories from those generated by the agent's policy, $\pi_\xi$, during imagined rollouts within our world model. The policy and discriminator are optimized via a minimax game where the discriminator's output provides a dense reward signal. This process minimizes the Jensen-Shannon divergence between the expert and policy visitation distributions in the latent space, governed by the objective:

$$\min_{\pi_\xi} \max_{D_\psi} \quad \mathbb{E}_{\tau_E \sim \mathcal{D}_E}[-\log D_\psi(s, a)]$$
$$+ \mathbb{E}_{\tau_\pi \sim \pi_\xi}[-\log(1 - D_\psi(s, a))] \tag{5}$$

By training the policy entirely on imagined rollouts, this approach effectively capitalizes on the pre-trained dynamics to achieve high sample efficiency in the target imitation task.

For the offline RL paradigm, we address the critical challenge of learning from a fixed dataset where distributional shift can lead to catastrophic failures. We integrate our framework with the principles of Latent Offline Model-Based Policy Optimization (LOMPO) (Rafailov et al., 2021a). Our uncertainty-aware pre-trained world model is uniquely suited for this paradigm. The ensemble dynamics, a core component of HAUWM, provides a natural and robust mechanism for quantifying epistemic uncertainty through model disagreement. For APV (Seo et al., 2022) and ContextWM (Wu et al., 2023), we also equipped them with ensemble dynamics models. To mitigate the risk of the policy exploiting out-of-distribution states where the model is inaccurate, we incorporate a pessimism principle directly into the reward function for imagined trajectories. Specifically, the reward is penalized by the variance of the predictions across the dynamics ensemble heads, discouraging the policy from venturing into uncertain regions of the state space. The uncertainty-penalized reward $\tilde{r}(s_t, a_t)$ is formulated as:

$$\tilde{r}(s_t, a_t) = r(s_t, a_t) - \lambda u(s_t, a_t) \tag{6}$$

Here, $u(s_t, a_t)$ is the uncertainty measured as the variance of the ensemble's predictive distributions, and $\lambda$ is a hyperparameter controlling the degree of conservatism. This ensures that the policy learns

Table 4: The absolute episodic return values for the random agent, the expert, and all ablated versions on all tasks averaged by four seeds.

| Task | Random | Expert | $\lambda = 10.0$ | $\lambda = 10^{-1}$ | $\lambda = 10^{-2}$ | w/o HCU | HAUWM |
|---|---|---|---|---|---|---|---|
| Dial Turn | $687 \pm 189$ | $4232 \pm 270$ | $3417 \pm 177$ | $\mathbf{3523 \pm 355}$ | $3381 \pm 248$ | $3258 \pm 709$ | $3225 \pm 281$ |
| Lever Pull | $239 \pm 215$ | $4913 \pm 379$ | $3838 \pm 234$ | $3978 \pm 467$ | $3791 \pm 327$ | $3725 \pm 935$ | $\mathbf{4022 \pm 288}$ |
| Door Open | $277 \pm 158$ | $4637 \pm 478$ | $3634 \pm 218$ | $3765 \pm 436$ | $3591 \pm 305$ | $3609 \pm 872$ | $\mathbf{3811 \pm 502}$ |
| Drawer Close | $221 \pm 198$ | $4895 \pm 578$ | $3820 \pm 234$ | $3960 \pm 467$ | $3773 \pm 327$ | $3707 \pm 935$ | $\mathbf{4736 \pm 393}$ |
| Cheetah Run | $71 \pm 38$ | $821 \pm 44$ | $574 \pm 98$ | $589 \pm 45$ | $596 \pm 30$ | $544 \pm 75$ | $\mathbf{626 \pm 23}$ |
| Walker Run | $120 \pm 61$ | $932 \pm 54$ | $664 \pm 106$ | $680 \pm 49$ | $688 \pm 32$ | $632 \pm 81$ | $\mathbf{721 \pm 24}$ |
| Hopper Hop | $25 \pm 12$ | $478 \pm 101$ | $329 \pm 59$ | $338 \pm 27$ | $342 \pm 18$ | $310 \pm 45$ | $\mathbf{360 \pm 14}$ |
| Quadruped Run | $35 \pm 19$ | $795 \pm 96$ | $544 \pm 99$ | $559 \pm 46$ | $567 \pm 30$ | $514 \pm 76$ | $\mathbf{597 \pm 23}$ |
| Push Green | $12 \pm 5$ | $378 \pm 77$ | $235 \pm 33$ | $232 \pm 18$ | $250 \pm 26$ | $213 \pm 33$ | $\mathbf{272 \pm 18}$ |
| Open Slide | $7 \pm 3$ | $177 \pm 55$ | $111 \pm 15$ | $109 \pm 9$ | $118 \pm 12$ | $101 \pm 15$ | $\mathbf{128 \pm 9}$ |

effective behaviors while remaining confined to the regions of the state space well-supported by the offline data. We choose the "medexp" dataset for each task as the offline dataset, a combination of medium replay and expert demonstrations.

### E.2 DETAILED EPISODIC RETURNS

To facilitate a clear and intuitive comparison, we report all performance metrics in the ablation (RQ2) and extension (RQ4) experiments as normalized returns. These returns are normalized for each task based on its corresponding random and expert performance. The absolute return values for the random agent, the expert, and all ablated versions are detailed in table 4.

### E.3 UNCERTAINTY COLLAPSE IN STANDARD RSSM-BASED BASELINES

To provide quantitative evidence that prevailing methods suffer from "false certainty", we conducted the following controlled experiment. We retrained the APV (Seo et al., 2022) and ContextWM (Wu et al., 2023) and replaced their single deterministic dynamics heads with an ensemble of $M = 7$ independent Gaussian heads with the same architecture as used in HAUWM, which ensures a strictly fair comparison among all methods. We then sampled 5,000 random frames from the Something-Something-v2 pre-training dataset (the same dataset used for all methods in our study). For each frame, we performed open-loop rollouts with horizons $k \in \{1, 2, 3, 4, 5\}$ steps and measured the dynamics uncertainty:

$$\text{Uncertainty}(k) = -\frac{1}{M-1} \sum_{i=1}^{M} \left( \mu_{\theta_i}(s_t, \Delta t_k^e) - \bar{\mu}_{t+k} \right)^2 \tag{7}$$

The results are shown in fig. 1 (b) (orange and green curves for APV and ContextWM, yellow curve for HAUWM). Both baselines exhibit near-zero growth in disagreement as the horizon increases, confirming that the standard single-step ELBO objective actively suppresses representation of stochasticity and induces deterministic, low-variance predictions even when the model architecture is capable of expressing uncertainty (i.e., when given ensemble heads). In contrast, HAUWM shows the desired monotonic increase, validating that the HCU loss is directly responsible for the calibrated growth of predictive uncertainty over time. This controlled experiment provides clear empirical evidence of the claimed defect in baseline methods and rigorously justifies the introduction of the HCU loss.

### E.4 SCALING FUNCTION IN THE HORIZON-CALIBRATED UNCERTAINTY LOSS

We adopt linear scaling $k$ in the HCU loss because it most directly encodes the physically intuitive prior that predictive uncertainty accumulates linearly with time—a simple, stable, and effective inductive bias for real-world dynamics. To confirm this design choice, we compared it against two alternatives: exponential scaling $e^k$ and periodic scaling $\left| \sin\left( k \cdot \frac{2\pi}{K_{\max}} \right) \right|$, where $K_{\max} = 5$ is the

maximum horizon used in pre-training. Performance on four representative tasks (mean $\pm$ std over four seeds) is reported in table 5.

Table 5: Ablation on the scaling function in the HCU loss (eq. (3)). Linear scaling achieves the best performance and training stability across all tasks.

| Scaling function | Cheetah Run | Hopper Hop | Dial Turn | Drawer Close |
|---|---|---|---|---|
| $k$ (linear, ours) | **626 ± 23** | **360 ± 14** | **3225 ± 281** | **4736 ± 393** |
| $e^k$ (exponential) | 587 ± 42 | 341 ± 27 | 3102 ± 156 | 4265 ± 349 |
| $\left|\sin\left(k \cdot \frac{2\pi}{K}\right)\right|$ | 421 ± 87 | 104 ± 55 | 2106 ± 333 | 3089 ± 796 |

Linear scaling consistently outperforms both alternatives. The exponential variant causes uncertainty to grow too aggressively, which disproportionately penalizes long-horizon predictions and destabilizes optimization, particularly in tasks with extended temporal dependencies. The periodic variant violates the monotonicity assumption and produces oscillating uncertainty signals that confuse the dynamics learning. These results validate linear scaling as the optimal balance between physical plausibility and training robustness.

### E.5 THE EFFECT OF THE NUMBER OF ENSEMBLE DYNAMICS HEADS

The ensemble size $M$ controls the trade-off between predictive diversity and mean accuracy. When $M = 1$, the disagreement term in eq. (3) vanishes, making the model equivalent to the "w/o HCU" ablation in table 1 of the main paper. We evaluated $M \in \{1, 3, 5, 7, 9\}$ on the same four tasks (mean $\pm$ std over four seeds). Results are shown in table 6.

Table 6: Ablation on ensemble size $M$. Performance peaks at $M = 7$, confirming that moderate ensemble diversity yields the best downstream control performance.

| Number of heads | Cheetah Run | Hopper Hop | Dial Turn | Drawer Close |
|---|---|---|---|---|
| M=9 | 579 ± 28 | 342 ± 22 | **3377 ± 298** | 4522 ± 299 |
| M=7 (ours) | **626 ± 23** | **360 ± 14** | 3225 ± 281 | **4736 ± 393** |
| M=5 | 589 ± 43 | 379 ± 55 | 3113 ± 257 | 4411 ± 330 |
| M=3 | 550 ± 94 | 327 ± 49 | 3177 ± 294 | 4087 ± 498 |
| M=1 (w/o HCU) | 544 ± 75 | 310 ± 45 | 3258 ± 709 | 3707 ± 935 |

Performance improves steadily from $M = 1$ to $M = 7$ as greater ensemble diversity better captures multi-modal futures. Beyond $M = 7$, marginal returns diminish and training becomes slightly noisier without meaningful gains, confirming $M = 7$ as a sweet spot that balances representational capacity, calibration quality, and computational efficiency.

