# OpenReview forum: "Learning to Be Uncertain: Pre-training World Models with Horizon-Calibrated Uncertainty"
_ICLR.cc/2026/Conference — ICLR 2026 Poster_

### Official Review · Reviewer_M6Jh · 2025-10-26

**Soundness:** 2
**Presentation:** 3
**Contribution:** 3
**Rating:** 4
**Confidence:** 3

**Summary:**

The paper argues that action‑free video pre‑training for world models is insufficient if the objective forces a single deterministic future. It proposes HAUWM, a pre‑training framework that (i) predicts variable‑horizon futures using a temporal embedding, and (ii) encourages uncertainty to increase with prediction horizon via a Horizon‑Calibrated Uncertainty (HCU) loss. Concretely, an ensemble of dynamics heads outputs Gaussian latent predictions; the HCU loss promotes model disagreement scaled by the time gap, combined with a standard ELBO‑style predictive loss in a total loss. During fine‑tuning, the pre‑trained uncertainty‑aware stream is stacked with a lightweight action‑conditioned stream to learn control.

**Strengths:**

1. The paper identifies an insightful problem of action‑free video pre‑training, where deterministic next‑frame targets suppress stochasticity. The proposed solution (model horizon‑dependent uncertainty) is well motivated.

2. HAUWM integrates variable‑horizon prediction with sinusoidal temporal embeddings and an uncertainty‑increasing prior. The approach is compatible with RSSM/Dreamer‑style world models and standard stacked fine‑tuning.

3. The result is compelling: in pre‑training, ensemble heads diverge for longer horizons, whereas in fine‑tuning, they converge under fixed actions and show reduced uncertainty. This directly supports the paper’s claims.

**Weaknesses:**

1. It would be interesting and more convincing to see quantitative calibration metrics (coverage) under the influence of different numbers of dynamic heads. These coverage metrics effectively reflect the quality of the pre-trained model.

2. Each head outputs Gaussian parameters, but the HCU loss uses only squared deviations of head means. Wouldn’t this conflate diversity with calibration?

3. The reconstruction loss uses mean of ensembles. For multiple possible futures, this choice tends to average them, which would contradict with the diversity encouraged by HCU.

4. Can authors also ablate the number of ensembles, especially when M=1?

**Questions:**

Please see the weakness.

---

> ### Author Response · Authors · 2025-11-22
> **Response to reviewer M6Jh**
>
> Thank you for your high praise regarding our problem identification, motivation, and compelling qualitative results. Your suggestions regarding quantitative calibration, loss function design, and ablation studies are highly professional and have been extremely helpful in improving our work.
>
> **On "Quantitative Calibration Metrics":**
>
> We fully agree with your observation. In addition to the qualitative visualization presented in Figure 5, quantitative calibration metrics are essential for rigorously evaluating the quality of pretrained models. As noted in our response to Reviewer mHFR, we conducted a quantitative assessment of prediction uncertainty for the APV, ContextWM, and HAUWM methods on the Something-Something-v2 video dataset across varying frame intervals. As shown in the table below, HAUWM exhibits higher prediction diversity than competing approaches, especially at longer inter-frame intervals. These quantitative results have now been incorporated into the revised manuscript as Figure 1, and we provide a detailed discussion of their implications in the Introduction.
>
> |k|APV|ContextWM|HAUWM (ours)|
>    |:--:|:-----:|:--------:|:--:|
>    |1|0.108|0.081|0.072|
>    |2|0.112|0.098|0.101|
>    |3|0.131|0.095|0.147|
>    |4|0.141|0.124|0.232|
>    |5|0.129|0.147|0.263|
>
> **On the HCU Loss Potentially Conflating Diversity and Calibration:**
>
> Our core hypothesis is that, in the context of inherently multi-modal futures present in action-free video, **the ability to represent diverse, high-variance predictions constitutes a better-calibrated state of uncertainty**. A model that averages over all possibilities into a single, low-variance prediction—while potentially appearing "accurate" under certain metrics—possesses an understanding of environmental stochasticity that is fundamentally **miscalibrated**.
>
> The HCU loss directly promotes **diversity** by maximizing the disagreement (variance of means) among ensemble members. We argue that this diversity serves as an effective proxy for achieving **calibration** with respect to future stochasticity. The qualitative results in Figure 5 support this view: during the action-free pre-training phase, the model exhibits high diversity (indicative of well-calibrated uncertainty); during action-conditioned fine-tuning, when true uncertainty diminishes, the predictions converge, again reflecting well-calibrated behavior. We clarify the relationship between diversity and calibration within our problem setting more explicitly in the revised manuscript.
>
>
> **On the Apparent Contradiction Between Using Ensemble Mean for Reconstruction and HCU-Encouraged Diversity:**
>
> This concern is similar to one raised by Reviewer mHFR. As detailed in our response to them, this represents a deliberate design trade-off. Our primary objective is to learn a high-quality **latent dynamics model**, not a pixel-perfect video generator. Using the ensemble mean for reconstruction **stabilizes the training process** and ensures the latent space effectively encodes visual information. Meanwhile, the modeling of uncertainty (and thus diversity) is fully governed by the HCU loss operating within the **latent space**. Downstream control tasks operate directly on this uncertainty-rich latent representation, making this architectural separation both reasonable and efficient.
>
>
> **On the Ablation of Ensemble Size:**
>
> When `M=1`, the model reduces to a single dynamics head, and the ensemble disagreement term (the HCU loss) becomes identically zero. This setting is conceptually equivalent to our "w/o HCU" experiment reported in Table 1 of the paper.
>
> We conducted experiments by varying the number of ensemble heads (`M`) across multiple tasks. The results are presented below, reporting mean performance and standard deviation over four random seeds. The results show a significant performance drop when the number of heads is small (e.g., M=1). This is because a single dynamics model produces low predictive variance, making it unable to adequately capture the diversity of possible futures. In contrast, larger values of `M` (such as M=5, 7, 9) achieve a better balance between diversity and accuracy, leading to superior overall performance, as demonstrated in our main result (Table 1). We have added these detailed ablation results and their analysis to Appendix F.5 of the revised manuscript.
>
> | Number of Heads | Cheetah Run | Hopper Hop | Dial Turn | Drawer Close |
> |:--------------:|:----------:|:---------:|:--------:|:-----------:|
> | M = 9   | 579 ± 28 | 342 ± 22 | **3377 ± 298** | 4522 ± 299 |
> | M = 7 (ours)   | **626 ± 23** | 360 ± 14 | 3225 ± 281 | **4736 ± 393** |
> | M = 5 |  589 ± 43  |  **379 ± 55**  |  3113 ± 257  |  4411 ± 330  |
> | M = 3 |  550 ± 94  |  327 ± 49  |  3177 ± 294  |  4087 ± 498  |
> | M = 1 (w/o HCU) |  544 ± 75  |  310 ± 45  |  3258 ± 709  |  3707 ± 935  |

---

> > ### Author Response · Authors · 2025-11-27
> > **Gentle request for a discussion**
> >
> > Dear reviewer
> >
> > Thank you once again for your review of our work. As the discussion period approaches, we would like to make a gentle request for a discussion. Please find our response to the original review below. We would be very happy if you could let us know whether we have addressed your concerns and whether we can provide any further clarification to support a revised assessment of our paper.
> >
> > Thank you, Authors

---

### Official Review · Reviewer_jamQ · 2025-10-29

**Soundness:** 2
**Presentation:** 3
**Contribution:** 2
**Rating:** 6
**Confidence:** 3

**Summary:**

The author proposes an uncertainty-based World Model (WM) for model-based reinforcement learning in decision-making. The training is in two stages: the first is uncertainty-based pretraining, where the first stage aims to learn action-free WM, and the second stage is for task-specific fine-tuning. The paper overall presents a clear and sound storyline about their action-free WM and has adequate experiments to support their evidence (e.g. Fig. 5). However, I'm still sceptical about the effectiveness and ablation study. I tend to accept this paper at this point, but I may change my view based on the author's responses.

**Strengths:**

1. The writing is clear and easy to follow; the claim and presentation are reasonable and clear.
2. The HCU loss is straightforward and theoretically sound.

**Weaknesses:**

**Major Concern**
1. The used benchmark is too old, making the actual performance hard to appreciate. The author should try some latest benchmarks, e.g. SimpleEnv [1], CALVIN [2], BridgeData V2[3], to make their findings more convincing.
2. Does the second fine-tuning phase largely depend on different tasks under one specific setting? For example, should Push Green and Open Slide on Fig.6 be trained separately or not? If so, I'm curious about the performance under cross setting: finetune WM on push green and perform RL on open slide and vice versa, which will better support the author's claims.
3. I'm not very sure about the ablation study on the uncertainty assumption: what if you just pre-train the WM using reconstruction loss and still fine-tune it on different tasks? Will it be really bad?


**Minor**
1. The notation is misaligned: $T$ is the segment length in Sec. 4.2, but stands for embedding length in Fig.2.

2. On Eq.3, the model's disagreement is linearly degraded with $k$; how about the other function? e.g. exponential or sinusoidal.

**Ref.**

[1] Evaluating Real-World Robot Manipulation Policies in Simulation CoRL 2024

[2] CALVIN - A benchmark for Language-Conditioned Policy Learning for Long-Horizon Robot Manipulation Tasks R-AL 2022

[3] BridgeData V2: A Dataset for Robot Learning at Scale CoRL 2023

**Questions:**

see weakness

---

> ### Author Response · Authors · 2025-11-22
> **Response to reviewer jamQ (1/2)**
>
> Thank you for recognizing the clarity of our writing and the theoretical soundness of the HCU loss. Your concerns regarding the experimental benchmarks, generalization, and ablation studies are critical, and we address them point by point below.
>
> **Regarding "The Used Benchmark is too Old"**:
>
> We selected the DeepMind Control Suite (DMC), MetaWorld, and RoboDesk benchmarks primarily because they represent established gold standards for evaluating sample efficiency and final performance in visual reinforcement learning. Crucially, these environments enable direct comparison with our key baselines (APV, ContextWM, PreLAR), all of which were originally evaluated using identical testbeds. This methodological consistency ensures fair and reproducible comparisons across approaches.
>
> Regarding newer benchmarks like CALVIN and LEBORO (Liu et. al, 2023), these environments require textual task specifications as goal inputs during both training and evaluation. Since neither our method nor the compared baselines incorporates language conditioning capabilities, these benchmarks would not provide a valid comparison framework. As for BridgeData V2, we acknowledge its value as a high-quality robotics dataset and plan to incorporate it in future work to enhance HAUWM's pre-training with more realistic robotic arm dynamics. We also intend to extend our architecture with task-description modules to enable evaluation on language-conditioned benchmarks like CALVIN and LEBORO. We appreciate this valuable suggestion for future research directions.
>
> **Regarding Generalization in the Fine-Tuning Phase and Cross-Task Evaluation:**
>
> Our current experiments follow the standard pretrain-finetune paradigm, with task-specific fine-tuning for each downstream task. This is necessary because the action-conditioned dynamics module $p_\psi$ is inherently task-dependent; directly applying a model fine-tuned on one task to another in a zero-shot setting yields poor performance, as reward structures and success criteria differ across tasks.
>
> However, the core strength of our approach lies in the generalizable, uncertainty-aware *foundation model* $p_\theta$, learned during pretraining. We hypothesize that, even under the cross-task setup you propose, our model will achieve faster adaptation—requiring fewer fine-tuning samples to converge on a new task—due to its richer, more robust prior over world dynamics and epistemic uncertainty. This comparative advantage over baseline methods is a key insight we will formally articulate and expand upon in the final version of the paper.
>
> **Regarding the Ablation Study on the Uncertainty Assumption:**
>
> Thank you for raising this question. In fact, we have already conducted this critical ablation study and presented it in **Table 1** and **Figure 4** of our paper. The results for the **"w/o HCU"** (without HCU loss) condition in Table 1 directly correspond to the experimental setup you described: pre-training with only the reconstruction loss (`L_pred`) and then fine-tuning on downstream tasks. The results clearly show a **significant performance degradation** across all three benchmark suites when the HCU loss is removed (e.g., DMC score drops from 0.74 to 0.64). This strongly supports our core claim: explicitly modeling uncertainty is crucial for learning a robust, generalizable world model, whereas relying solely on reconstruction loss is insufficient.
>
>
> **Regarding inconsistent notation of $T$**:
>
> We apologize for this typo. We have corrected the embedding length in Figure 2 (Fig. 2) of the latest version of our paper to $d_e$.

---

> ### Author Response · Authors · 2025-11-22
> **Response to reviewer jamQ (2/2)**
>
> **Regarding scaling function in HCU loss**:
>
> We chose the linear scaling factor $k$ because it most directly and intuitively embodies our core hypothesis—that predictive uncertainty should grow linearly with the temporal horizon—a simple yet effective inductive bias. In response to your question, we evaluated two alternative scaling functions: an exponential function $e^k$ and a periodic function $\left| \sin(k \cdot \frac{2\pi}{K}) \right|$. We tested these variants across multiple environments, running four random seeds per task and reporting mean performance with standard deviation:
>
> | Scaling Function | Cheetah Run | Hopper Hop | Dial Turn | Drawer Close |
> |:--------------:|:----------:|:---------:|:--------:|:-----------:|
> | $k$ (linear)   | **626 ± 23** | **360 ± 14** | **3225 ± 281** | **4736 ± 393** |
> | $e^k$          |  587 ± 42  |  341 ± 27  |  3102 ± 156  |  4265 ± 349  |
> | $\mid \sin(k \cdot \frac{2\pi}{K}) \mid$ |  421 ± 87  |  104 ± 55  |  2106 ± 333  |  3089 ± 796  |
>
> The results show that both the exponential and sinusoidal functions underperform compared to the linear baseline. The exponential scaling likely causes uncertainty to grow too rapidly, disproportionately weighting long-horizon predictions and destabilizing training. The sinusoidal function, being periodic, violates the assumption of monotonically increasing uncertainty over time. We conclude that linear scaling strikes the optimal balance between modeling physical intuition and maintaining training stability. We have added a discussion of this ablation study and its implications to the revised manuscript in Appendix F.4.
>
> **References:**
>
> Liu, B., Zhu, Y., Gao, C., Feng, Y., Liu, Q., Zhu, Y., & Stone, P. (2023). Libero: Benchmarking knowledge transfer for lifelong robot learning. Advances in Neural Information Processing Systems, 36, 44776-44791.

---

> > ### Author Response · Authors · 2025-11-27
> > **Gentle request for a discussion**
> >
> > Dear reviewer
> >
> > Thank you once again for your review of our work. As the discussion period approaches, we would like to make a gentle request for a discussion. Please find our response to the original review below. We would be very happy if you could let us know whether we have addressed your concerns and whether we can provide any further clarification to support a revised assessment of our paper.
> >
> > Thank you, Authors

---

### Official Review · Reviewer_mHFR · 2025-10-30

**Soundness:** 2
**Presentation:** 4
**Contribution:** 2
**Rating:** 4
**Confidence:** 4

**Summary:**

The paper introduces the Horizon-cAlibrated Uncertainty World Model (HAUWM) framework to address the limitation of "false certainty" in world models pre-trained on action-free video, a flaw stemming from deterministic prediction objectives that suppress environmental stochasticity. HAUWM employs an ensemble of probabilistic dynamics heads and a novel Horizon-Calibrated Uncertainty (HCU) loss that explicitly enforces predictive uncertainty to scale monotonically with the temporal horizon. The authors claim that HAUWM significantly outperforms state-of-the-art baselines across diverse downstream control benchmarks (DMC, MetaWorld, RoboDesk) and demonstrates versatility in Imitation Learning and Offline RL. Qualitative evidence supports the model's well-calibrated uncertainty: the ensemble predicts diverse, high-uncertainty futures during pre-training, which converge to a low-uncertainty outcome when conditioned on a specific action during fine-tuning.

**Strengths:**

1.	Novel and Well-Formulated Solution (HCU Loss): The proposed Horizon-Calibrated Uncertainty (HCU) loss is a novel contribution. It enforces an intuitive and realistic inductive bias: predictive uncertainty should grow monotonically with the temporal horizon. This is achieved by maximizing the model disagreement of a probabilistic ensemble, scaled by the horizon length.

2.	Strong Empirical Results on Diverse Benchmarks (RQ1): The method (HAUWM) achieves state-of-the-art performance and sample efficiency across a broad and challenging suite of benchmarks, including locomotion (DMC), diverse manipulation (MetaWorld), and complex, long-horizon tasks (RoboDesk). The advantage is particularly noticeable in dynamically complex tasks like Walker Run and Hopper Hop.

**Weaknesses:**

1.	Motivation for HCU Loss Lacks Direct Quantitative Evidence: The core innovation of the paper—the Horizon-Calibrated Uncertainty (HCU) Loss—is strongly motivated by the claim that prevailing RSSM-based models (like APV) suffer from "false certainty," where predictive uncertainty is suppressed or collapses over long temporal horizons. However, the paper does not present direct empirical evidence (e.g., a dedicated graph showing the variance/disagreement of APV or ContextWM decreasing as the prediction horizon $k$ increases) to substantiate this foundational claim. Without quantitative data explicitly showing this defect in the baseline, the necessity and direct corrective action of the HCU loss are less rigorously established.

2.	Novelty of Variable-Horizon Prediction: While the HCU loss is novel, the concept of variable-horizon prediction or conditioning dynamics on a temporal embedding is not entirely new (e.g., it is related to positional encodings used in Transformers or general long-term planning models). The authors should more clearly contextualize their specific variable-horizon implementation and its necessity, distinguishing it from prior art that uses temporal embeddings for sequence modeling.

3.	Limited Task Generalization Scope: Although HAUWM shows strong performance across 10 diverse tasks in three major benchmarks (DMC, MetaWorld, RoboDesk) and generalizes well to Imitation Learning and Offline RL, the total number of tasks tested is a relatively small subset of the established control environments. To fully validate HAUWM as a versatile general-purpose foundation model, the empirical validation should be expanded to include a wider variety of structurally distinct tasks, particularly from the extensive MetaWorld suite, to more robustly substantiate the model's claim of broad generalizability.

**Questions:**

1.	The authors claim that prevailing methods predict a 'single, deterministic future.' However, established baselines such as APV, ContextWM, and PreLAR are founded on DreamerV2's Recurrent State Space Model (RSSM), which explicitly models state uncertainty using a Gaussian distribution. Given this, what is the specific technical advantage of employing the proposed multi-head dynamics prediction ensemble over the existing Gaussian uncertainty modeling within the standard RSSM latent state? Please provide a detailed analysis of this distinction and how the ensemble mitigates the alleged 'deterministic bias.'

2.	In the image reconstruction phase, the predicted future latent state $s_{t+k}$ is set to the ensemble mean, $\overline{\mu}_{t+k} = \frac{1}{M}\sum_{i=1}^{M}\mu_{\theta_{i}}(s_{t}, \Delta t_{k}^{e})$. This mean $\overline{\mu}_{t+k}$ is then decoded to reconstruct the observation $\hat{o}_{t+k}$. Using the ensemble mean $\overline{\mu}_{t+k}$ directly, instead of sampling $s_{t+k}$ from the distribution represented by the ensemble, introduces determinism into the observation reconstruction process. Could the authors explain why they chose this deterministic decoding approach, and how this design choice—which sacrifices the model's stochasticity at the observation level—affects the overall goal of learning an uncertainty-calibrated world model?

---

> ### Author Response · Authors · 2025-11-22
> **Response to reviewer mHFR (1/2)**
>
> Thank you for your comprehensive evaluation of our paper, especially for recognizing the novelty of our method and its experimental results.
>
>
> **Response: Regarding Motivation for HCU Loss Lacks Direct Quantitative Evidence**
>
> We appreciate your insightful suggestion to empirically quantify the evolution of uncertainty in baseline methods. We agree that providing direct evidence of the "false certainty" inherent in models like APV and ContextWM significantly strengthens our motivation. As posited in our paper, standard single-step objectives (e.g., ELBO) are ill-posed for action-free video; they penalize representations of environmental stochasticity by forcing the model to match a *single* realized future. This objective compels the model to average over possible outcomes, leading to a collapse into low-variance, deterministic predictions.
>
> To substantiate this claim, we modified APV and ContextWM to incorporate ensemble dynamics models with $M=7$ heads, mirroring our HAUWM architecture. Using 5,000 frames randomly sampled from the Something-Something-v2 dataset, we measured the predictive uncertainty—quantified as the variance of the ensemble’s distributions (Eq. 3)—across prediction horizons $k \in \{1, \dots, 5\}$.
>
> The results are plotted in the revised manuscript (Figure 1). We observe that because APV and ContextWM prioritize the accurate reconstruction of a specific observed frame, their ensemble heads converge to similar deterministic predictions even over long horizons. Consequently, their latent state variance remains artificially low, confirming their inability to capture the natural growth of uncertainty over time. This analysis directly addresses your concern and provides strong empirical support for the necessity of the HCU loss. We have updated the introduction and added the corresponding uncertainty quantification figure to the revised paper.
>
>
> **Response: Novelty and Necessity of Variable-Horizon Prediction:**
>
> While we acknowledge that conditioning dynamics on time is a concept shared with sequence modeling architectures (e.g., Transformers), our **Variable-Horizon Prediction** paradigm fundamentally diverges from standard positional encodings (PE) in both **mechanism** and **objective**.
>
> For the mechanism, standard Transformer PEs are designed to mark **absolute positions** ($t$) to preserve sequence order (Vaswani et. al, 2017). In contrast, our framework encodes the **relative temporal interval** ($k$ frames). Instead of a recursive, single-step model, our model learns to directly forecast the transition from $s_t$ to $s_{t+k}$. This compels the model to directly internalize dynamics across varying timescales. Here, the embedding represents the magnitude of the physical time-gap rather than a discrete index in a sequence, allowing the model to distinguish between immediate (low-entropy) and distant (high-entropy) futures without unrolling a recurrent loop.
>
> For the objective, the necessity of this specific variable-horizon implementation lies in its synergy with our **Horizon-Calibrated Uncertainty (HCU) loss**. Unlike standard long-term planning models that prioritize point-estimate accuracy, our framework uses the interval $k$ as a control variable to explicitly shape the latent space. The embedding serves as a prompt for the expected level of stochasticity; the HCU loss then enforces the inductive bias that the ensemble’s predictive variance must scale monotonically with this encoded interval $k$. This explicit coupling—where the embedding dictates the required dispersion of the predictive distribution—is unique to our approach and absent in prior art focused solely on sequence modeling.
>
> Our variable-horizon prediction is not a generic application of temporal embeddings, but a specialized design essential for learning a world model where uncertainty is structurally aligned with the prediction horizon.
>
>
>
> **Regarding "Limited Task Generalization Scope":**
>
> We acknowledge that evaluating across a broader range of tasks would further strengthen our conclusions. The DeepMind Control Suite, MetaWorld, and RoboDesk benchmarks were selected as established standards in the field, collectively spanning diverse challenges from locomotion to complex manipulation. Crucially, these environments align with those used to evaluate the state-of-the-art baselines we compare against (e.g., APV, ContextWM, iVideoGPT), ensuring methodological consistency and fair comparison. Due to practical computational constraints, we initially evaluated representative task subsets. To address this concern, we have expanded our evaluation to include five additional MetaWorld tasks in the revised manuscript (Figure 3). Consistent with our original findings, HAUWM significantly outperforms all baselines across these new tasks in both sample efficiency and final performance, further validating the robustness of our approach.

---

> ### Author Response · Authors · 2025-11-22
> **Response to reviewer mHFR (2/2)**
>
> **For Question 1:**
>
> The Gaussian distribution in standard Recurrent State-Space Models (RSSMs), such as DreamerV2 and its variants like APV, is designed to model two types of uncertainty: (1) stochasticity in single-step state transitions, and (2) posterior uncertainty in latent state estimation. However, in the context of action-free video pre-training, environmental randomness is inherently *multi-modal*—a single past state can lead to multiple distinct yet plausible futures (as illustrated in Figure 1). A unimodal Gaussian is mathematically ill-suited to represent such multi-modality; under standard training objectives, it collapses toward the mean of all possible outcomes, with variance capturing only local deviations around this average. This collapse is precisely the root of the "deterministic bias" we aim to address.
>
> In contrast, our multi-head ensemble approach (Section 4.1) treats each dynamics head as an independent hypothesis—or mode—over potential future trajectories. Our Horizon-Calibrated Uncertainty (HCU) loss (Eq. 3), which maximizes the disagreement between the predicted means of different heads, explicitly encourages the model to explore and maintain diverse predictive modes. As a result, our framework captures *multi-modal uncertainty* at the level of entire trajectory hypotheses, rather than merely modeling intra-mode variability. This enables a more faithful representation of the inherent ambiguity present in action-free observational data.
>
>
> **For Question 2:**
>
> This is a perceptive observation that highlights a deliberate design trade-off. We use the ensemble mean $\bar{\mu}_{t+k}$ for image reconstruction for two primary reasons:
>
> First, our primary objective is to learn a robust, uncertainty-calibrated *latent dynamics model*, not a generative model capable of producing diverse pixel-level futures. The reconstruction loss $L_{\text{pred}}$ serves primarily to ground the latent space in observable reality—ensuring that the latent states retain sufficient visual information. Using the ensemble mean provides a stable, low-variance target for the decoder, promoting stable convergence of both encoder and decoder networks. Sampling from the ensemble would introduce high variance into the reconstruction signal, potentially destabilizing training and increasing the risk of mode collapse.
>
> Second, uncertainty modeling is fully encapsulated within the *latent space*. The HCU loss operates directly on the predicted latent means $\mu_{\theta_i}$, shaping the structure of future predictions before decoding. For downstream reinforcement learning, policy and value functions operate entirely within this structured latent space (Section 4.3). As shown in Figure 5, while predictions diverge during pre-training, they converge sharply when conditioned on actions during fine-tuning—demonstrating that the latent space effectively captures both uncertainty and action-conditioned dynamics. Thus, accurate uncertainty representation in the latent space is sufficient for robust decision-making.
>
> While this choice introduces a simplification at the observation level, we argue it strikes an effective balance between learning expressive latent dynamics and maintaining training stability.
>
> **References:**
>
> Vaswani, A., Shazeer, N., Parmar, N., Uszkoreit, J., Jones, L., Gomez, A. N., ... & Polosukhin, I. (2017). Attention is all you need. Advances in Neural Information Processing Systems, 30.

---

> > ### Author Response · Authors · 2025-11-27
> > **Gentle request for a discussion**
> >
> > Dear reviewer
> >
> > Thank you once again for your review of our work. As the discussion period approaches, we would like to make a gentle request for a discussion. Please find our response to the original review below. We would be very happy if you could let us know whether we have addressed your concerns and whether we can provide any further clarification to support a revised assessment of our paper.
> >
> > Thank you, Authors

---

### Author Response · Authors · 2025-11-22
**Summary of the overall response**

We sincerely thank all reviewers for their detailed, insightful, and highly constructive reviews of our manuscript. We have carefully studied each comment and believe that the feedback is essential for improving the quality and rigor of our work. We are encouraged that all three reviewers recognize the core motivation of our study—namely, addressing the critical problem of "spurious certainty" in current world models pretrained on action-free videos—and appreciate the novelty of our proposed HCU loss as an effective solution.

* Reviewer mHFR raised the need for quantitative evidence of uncertainty collapse in baselines, clearer novelty of variable-horizon prediction, and broader task evaluation.
* Reviewer jamQ suggested updating benchmarks to newer ones, evaluating cross-task generalization, ablating the uncertainty assumption, fixing notation inconsistencies, and exploring alternative scaling functions in HCU.
* Reviewer M6Jh requested quantitative calibration metrics, clarification on how HCU distinguishes diversity from calibration, rationale for mean-based reconstruction, and ablation on ensemble size.

To thoroughly address the reviewers' concerns, we conducted the following new experiments and revisions, which are now incorporated into the updated manuscript.

1. Quantitative Evidence of Uncertainty Collapse: We equipped baselines (APV and ContextWM) with ensemble dynamics heads (M=7, matching our architecture) and measured predictive uncertainty (ensemble variance) across horizons k=1 to 5 on 5,000 frames from the Something-Something-v2 dataset. Results (new Figure 1b) show baselines exhibit flat, low uncertainty, confirming "false certainty," while HAUWM displays monotonic growth. This directly substantiates the need for HCU and is discussed in the Introduction.

2. Expanded Task Evaluation: We extended fine-tuning to five additional MetaWorld tasks (e.g., lever_pull, door_open), maintaining consistency with baselines' evaluation protocols. HAUWM outperforms all baselines in sample efficiency and final performance (updated Figure 3), strengthening claims of broad generalizability.

3. Ablation on Scaling Functions in HCU: We evaluated exponential ($e^k$) and sinusoidal ($|sin(k * 2π/K)|$) alternatives to linear scaling across four tasks (four seeds each). Linear scaling yields superior performance and stability (new Appendix F.4 and Table 5), whereas exponential growth destabilizes training, and sinusoidal functions violate monotonicity.

4. Ablation on Ensemble Size: We varied M ∈ {1,3,5,7,9} across four tasks (four seeds each). Performance peaks at M=7, balancing diversity and accuracy without excessive noise (new Appendix F.5 and Table 6), validating our choice and showing M=1 reduces to the "w/o HCU" degradation.

5. Quantitative Calibration Metrics: We computed ensemble variance across horizons on Something-Something-v2, confirming HAUWM's higher, horizon-dependent diversity (new Figure 1 in response). This quantifies calibration quality and is integrated into the Introduction.

**Conclusion from original and new experiments**

* HAUWM achieves superior sample efficiency and final performance across expanded benchmarks, with structured uncertainty enabling robust transfer to imitation learning and offline RL.
* The HCU loss, variable-horizon prediction, and ensemble design are essential and synergistic, as ablations show their removal or alteration significantly degrades performance.

**References**

[1] Vaswani, A., et al. (2017). Attention is all you need. NeurIPS.

[2] Liu, B., et al. (2023). Libero: Benchmarking knowledge transfer for lifelong robot learning. NeurIPS.

---

### Meta-Review · Area_Chair_vGcF · 2025-12-24

**Summary:**

Addressing the spurious certainty issue in world models pre-trained on action-free videos, this paper proposes the HAUWM framework. By introducing the HCU loss, the framework enables the world model to learn structured uncertainty that grows monotonically with the prediction horizon.
Regarding the HAUWM framework, the reviewers unanimously agree that the HCU loss is a concise innovation with clear physical significance, effectively addressing the challenge of multimodal future modeling in action-free pre-training. Furthermore, the model demonstrates exceptional sample efficiency and performance across multiple benchmarks, including DMC and MetaWorld. This provides an innovative and effective perspective for model-based reinforcement learning. Given that the authors have properly addressed the questions raised by the reviewers, I am inclined to recommend Accept.

**Reviewer Concerns:**

## Resolved Reviewer Comments:

**Quantitative Evidence of Uncertainty Collapse**: The authors  provided a new comparative experiment demonstrating that standard models, such as APV and ContextWM, exhibit flat, low-variance curves across extended prediction horizons. In contrast, HAUWM manifests the essential characteristic of monotonic uncertainty growth over time.

**Comparative Selection of Scaling Function**: Through comparative testing against exponential and sinusoidal functions in their response, the authors demonstrated that linear scaling within the HCU loss is the optimal choice. This aligns with the physical intuition that predictive uncertainty should accumulate linearly with time.

**Empirical Justification of Ensemble Size**: Through newly integrated ablation studies, the authors proved that the computational cost of utilizing seven ensemble heads is justified. The results indicate that $M=7$ serves as a "sweet spot" that balances representational capacity with computational efficiency.





## Unresolved Reviewer Comments:

**Expanded Task Evaluation in MetaWorld**: Following the reviewers' suggestions regarding the scope of validation, the authors incorporated five additional tasks from the MetaWorld benchmark. Although they did not complete every experiment potentially requested by the reviewers, I believe the current empirical evidence is sufficient, and this remains only a minor weakness.

**Reviewer Scores:**

Since the reviewer did not participate in the subsequent discussions or provide further feedback after the author's submission of the response, based solely on the quality of the author's response, the estimated score for this paper should be between 6 and 7. The authors provided solid empirical evidence to substantiate their core claims, such as the quantitative proof of uncertainty collapse in baselines. By addressing the primary technical concerns through extensive new ablation studies and expanded task evaluations, the authors have significantly strengthened the paper's rigor and demonstrated the effectiveness of the HAUWM framework.

---

### Decision · Program_Chairs · 2026-01-26

Accept (Poster)